# Simulation of High-Performance Surface Plasmon Resonance Sensor Based on D-Shaped Dual Channel Photonic Crystal Fiber for Temperature Sensing

**DOI:** 10.3390/ma16010037

**Published:** 2022-12-21

**Authors:** Haoyu Wu, Yutong Song, Meng Sun, Qi Wang

**Affiliations:** College of Sciences, Northeastern University, Shenyang 110819, China

**Keywords:** surface plasmon resonance, photonic crystal fiber, mode coupling, temperature sensing

## Abstract

This paper presents and numerically analyzes a refractive index sensor based on side-polished D-shaped two-channel photonic crystal fiber (PCF) and surface plasmon resonance (SPR). The effects of pore duty ratio, polishing depth, and thickness of a Nano-Titania sensitizing layer on sensor performance are studied, and the sensor performance is analyzed and optimized. The results show that the sensitivity of the Nano-Titania sensitized sensor can reach 3392.86 nm/RIU and temperature sensitivity of the sensor is increased to 1.320 nm/K, and the amplitude sensitivity of the unsensitized sensor can reach 376.76 RIU^−1^. In addition, the influence of titanium dioxide layer on the mode field diameter of PCF fiber core is also studied. It is found out that the sensor with a 50 nm thick titanium dioxide film has a larger mode fiber diameter, and is more conducive to coupling with single-mode fiber. Our detailed results contribute to the understanding of SPR phenomena in hexagonal PCF and facilitate the implementation and application of SPR-PCF sensors.

## 1. Introduction

As technology develops, many effective optical fiber sensor technologies for detecting refractive index [1] have been developed, such as gratings [2] and fiber interferometers [3]. High sensitivity, light weight, anti-electromagnetic interference, and low production costs are a few of the benefits that optical fiber sensors have over traditional varieties. Photonic crystal fibers (PCF) have great qualities such evanescent wave intensity, low transmission loss, simple optical coupling, and huge tunable waveguide dispersion due to the strong refractive index contrast and sub-wavelength diameter [4]. Since the 1980s, surface plasmon resonance (SPR) optical sensors have been one of the most widely used techniques for simple and fast label-free detection [5,6]. In recent years, PCF-based SPR sensors have gained popularity due to their small size and low loss. Plasma waves are collective modes of electron oscillations in the conduction band at the interface of two materials, and are coherent with the incident oscillating electromagnetic field. Noble metal nanoparticles can excite surface plasmon resonance along metal-dielectric interfaces.

Recently, surface plasmon resonance technology has become a popular method for sensing and detecting information. Surface plasmon resonance (SPR) sensors have been widely used in chemical [7,8] and biochemical sensing [9], gas sensing [10], medical diagnosis [11], temperature monitoring [12], and food safety detection [13] for years. Total internal reflection of TM-polarized input light at the prism/metal film contacts results in evanescent wave propagation through the thin metal film. When the propagation constant of the hopping light along the interface is combined with that of the surface plasmon wave (SPW), which is confined at the interface and drops rapidly laterally, the resonant excitation of electrons will occur, accompanied by a sharp drop in reflectivity.

Applications of temperature monitoring [12] and environment have explored various SPR-based configurations [14], including SPR sensors based on prisms and SPR sensors based on optical fiber. Comparable to the surface plasmon resonance sensor, the Bloch surface wave sensor could adopt similar multilayer construction on the optical grating [15], fiber [16], and prism [17]. Esteban Gonzalez-Valencia designed and manufactured Bloch wave excitation platform based on few-layer photonic crystal deposited on D-shaped optical fiber [16]. Dalibor Ciprian’s team has designed a Bloch Surface Wave Resonance Based Sensors for monitoring ambient humidity [17]. SPR sensors have higher sensitivity and similar resolution to Bloch surface wave sensors. Additionally, fiber Bloch surface wave-based sensors are more prevalent in the infrared band. Traditional SPR sensors based on prisms are, however, limited in their practical uses by their huge size and expensive cost. Compared with the traditional prism-coupled SPR sensor, PCF has higher sensitivity and can effectively avoid the above defects. PCF fibers provide substantial advantages [18] over conventional fibers [19], including tunable birefringence, high confinement field, compact sensing, tunable dispersion, and the creation of single-mode light during propagation. Therefore, PCF-based sensors have higher sensitivity than traditional prism-coupled SPR sensors and can effectively avoid the defects of traditional sensors.

Plasmonic metals, including gold, silver, copper, and metal oxides, could ensure the enrichment of free electrons in the instantaneous field and are commonly used to excite SPR [20]. LSPR (localized surface plasmon resonance) is the result of a collective oscillation of free electrons in the conduction band of metallic nanoparticles (Au and Ag) (NPs). When coated with copper and certain metal oxide films, the sensor has low sensitivity and unclear resonance peak, making it difficult to identify the resonance wavelength. Silver is prone to oxidation, but the resulting silver oxide can lead to poor sensor performance [21]. Alkali metals [22] have been shown to excite SPWs with high sensitivity and clear resonance peaks, but due to their harsh preparation conditions, it is difficult to use alkali metals in practical sensors. By comparison, gold has excellent corrosion and oxidation compatibility, superior optical reactivity, and excellent chemical fixation. Compared with sensors coated with silver film, the sensors coated with gold film has sharper loss peak and smaller full width at half maximum, and, thus, are more conductive to the detection of the spectrometer [20]. 

There have been countless surface plasmon sensors built on side-polishing crystal fiber developed in recent years, some of which have seen widespread application. Rashed, Ahmed Nabih Zaki created a unique photonic crystal fiber structure that was tested in the refractive index range of 1.40 –1.45 RIU [23]. Amiri, I. S., designed a D-shaped dual-core photonic crystal fiber with a wavelength resolution of 1.25 × 10^−5^ RIU [24]. The research of the dual-core micro-channel fiber sensor’s properties in the visible and infrared bands by Bing, Pibin, and his team made significant improvements to the sensor’s capability to detect refractive indices [25]. Song, Li proposes a concept of an orthogonal-side polished microstructured optical fiber (MOF)-based surface plasmon resonance (SPR) sensor to implement simultaneous sensing for the temperature and refractive index [26]. Liu, Feng fabricated D-shape polished dual-core photonic crystal fiber with a polishing Angle of 60° and a polishing depth of 58.02 um with a sensitivity of 3580 nm/RIU [27]. Hossain, Md. Biplob offered a Quasi D-Shape dual core Photonic Crystal Fiber Surface Plasmon Resonance sensor, with maximum amplitude sensitivity of 230 RIU^−1^ and wavelength sensitivity of 15,000 nm/RIU [28]

In this paper, a high-sensitivity D-shaped dual-core PCF surface plasmon resonance (DD-PCF-SPR) sensor using gold as the outer surface plasmon material is studied. It can reliably quantify the change in temperature in a specific medium. In addition to the temperature dependence of the sensor film thickness and the metal-dielectric function, the dispersion of the fiber material and the temperature dependency of its refractive index are also taken into consideration in the theoretical model. For various polishing depths and duty ratios, the resonance spectra and coupling parameters of the polarized fundamental core mode and the surface plasmon polariton (SPP) mode are determined numerically.

## 2. Device Design and Theoretical Foundation

The pattern of the microstructured fiber is designed to be as simple and straightforward as possible. In contrast with the common commercial PCF, its structure is modified to have two separate optical channels at the center. Figure 1 depicts the geometrical schematic diagram of the PCF-SPR sensor. On the cladding, there are many layers of air holes, and the optical channel is divided into two parts by the central air hole, allowing for the transmission of the two light beams, respectively. The upper side of the PCF is polished, and the vertical distance from the polishing plane to the center of the fiber core is defined as h. The lattice pitch Λ is 7.9 μm, and the diameter of air hole is 3.95 μm. The exposed core plane is obtained after side polishing. The outer regions of the PCF are coated with a gold layer with variable thickness using vacuum magnetic sputtering. Since changes in temperature result in modifications to the PCF-SPR sensor elements, the phase matching conditions between both the core-guided and plasmonic modes may shift. The core is made of fused silica material. Using Sellmeier’s formula of refractive index [29], the change in refractive index with temperature can be calculated as follows.
(1)$$n_{\mathrm{silica}\;}^{2}=1.32+0.69\times 10^{-5}T+\frac{\left(0.79+0.24\times 10^{-4}T\right)\lambda^{2}}{\lambda^{2}-\left(0.01+0.58\times 10^{-6}T\right)}+\frac{\left(0.91+0.55\times 10^{-6}T\right)\lambda^{2}}{\lambda^{2}-100}$$
where *λ* represents the light’s wavelength and *T* represents the temperature of the fused silica.

The PDMS solution with temperature sensitive properties for the substance is chosen to be tested. There is no need to take the viscosity of the PDMS into account because we need to encapsulate the sensing region. Since PDMS is chemically stable and long-lasting at temperatures between −50 °C and 200 °C, it essentially fits the temperature range mentioned in this research. The following empirical formula illustrates the change in the refractive index of PDMS with temperature [30]:(2)$$RI=-4.5*10^{-4}*T+1.4176$$
where *T* is the temperature of the PDMS solution. Gold’s dielectric permittivity is obtained using Drude’s dispersive model [31,32].
(3)$$\varepsilon \left(\omega \right)=\varepsilon_{1}+i\varepsilon_{2}=\varepsilon_{\infty }-\frac{\omega_{p}^{2}}{\omega \left(\omega +i\omega_{c}\right)}$$
where $\varepsilon_{\infty }$ is the dielectric constant of gold at high frequencies. Consequently, ω represents the angular frequency. $\omega_{p}$ indicates the plasma frequency, while $\omega_{c}$ represents the collision frequency of metallic electrons.

Temperature influences both the plasma and collision frequency of metal electrons. On the basis of the volumetric effects, the plasma frequency may be expressed as [33]:(4)$$\omega_{p}=\omega_{p0}\exp\left(-\frac{T-T_{0}}{2}\times 3\gamma \right)$$
where $\omega_{p0}$ is the plasma frequency at $T_{0}$ and γ is the volume thermal expansion coefficient of gold.

The variation of the collision frequency ($\omega_{c}$) of metal electrons with temperature is more complicated than that of the plasma frequency. $\omega_{c}\;$ is influenced by two variables: phonon-electron scattering ($\omega_{cp}$) and electron-electron scattering ($\omega_{ce}$) [34]:(5)$$\omega_{c}=\omega_{ce}+\omega_{cp}$$

Electron scattering can be deduced based on Lawrence’s model of electron scattering [35]. In accordance with the Holstein phonon–electron scattering model, the phonon–electron scattering calculation formula [36,37]:(6)$$\begin{array}{r} \omega_{c}\left(T\right)=\frac{1}{6}\pi^{4}\frac{\Gamma \Delta }{\hbar E_{F}}\left[{\left(K_{B}T\right)}^{2}+{\left(\frac{\hbar \omega }{4\pi^{2}}\right)}^{2}\right]+\omega_{p}\left(T_{0}\right)\left[\frac{2}{5}+4{\left(\frac{T}{T_{D}}\right)}^{5}\int_{0}^{T_{D}/T}\frac{z^{4}dz}{e^{z}-1}\right] \end{array}$$
where $T_{D}$ is the Debye temperature, and the h, $E_{F}$, and $K_{B}$ denote the Planck constant, the Fermi energy of a metal electron, and the Boltzmann constant, respectively. According to the principle of thermal expansion and contraction, temperature changes also affect the thickness of the gold film. By correcting the thermal expansion coefficient, the relationship between the metal thickness and temperature can be obtained as [38]:(7)$$d_{Au}=d_{0}\left[1+\gamma \frac{1+\mu }{1-\mu }\left(T-T_{0}\right)\right]$$
where *d*_0_ is the thickness of the gold film at 298 K room temperature and μ is the Poisson’s number of the film material. Table 1 contains a comprehensive listing of all simulation-related parameters [39].

The propagation of the electric field mode in the optical fiber is depicted in Figure 2 after we re-simulated the simulation of the SMF and PCF splicing interface. 

Part of the light in the single-mode fiber that is visible light with an operating wavelength of 632.8 nm is reflected at the air interface, and the remaining light enters the two sensing channels of the dual-core PCF and gradually stabilizes. The requirement of the experiment is satisfied by the total transmittance of light passing through SMF–PCF–SMF being able to reach 10–20%.

## 3. Results and Discussion

In order to solve the complex propagation constants of the y-polarized core waveguide mode and the SPP mode, the full-vector finite element method is adopted under the condition of a cylindrical perfectly matched layer (PML). Simulations are conducted using COMSOL Multiphysics 5.6, a commercial software package.

### 3.1. Excitation of Surface Plasmon Resonance

The SPR detection principle is the interaction between the instantaneous field and electrons on the surface of the dielectric metal. At a particular wavelength, the real part of the effective mode refractive index of the two sets of core plasmons at the core and surface plasmons is equal. At the metal-core interface, the polarized mode couples to the surface plasmon mode due to the presence of an evanescent field. Upon witnessing the confinement loss of the sensor, the numerical performance of the plasmonic sensor is explored and characterized by the following formula [40]:(8)α(λ,na)(dB/cm)=8.686×2πλIm(neff)
where $\alpha \left(\lambda ,n_{a}\right)$ represents the radiation loss of the core mode. Im(neff) is the imaginary part of the effective refractive index, and *λ* is in centimeters. When the analyte RI is 1.35, the dispersion and confinement losses for the y-polarized fundamental core mode and the SPP mode are given. As the wavelength increases, the confinement loss first increases and then decreases. The core guide mode and plasma mode meet the phase matching conditions, resulting in energy loss. Thus, part of the core transfers energy to the metal surface to produce plasma oscillations. Due to the reaction of plasma oscillations to the leaky core mode, the wavelength at which the confinement loss of the y-polarized fundamental mode reaches a loss maximum shift to the red from the dispersion intersection.

Detailed observations in Figure 3 show that the loss of the core guided mode can reach 88.08 dB/cm at 587 nm resonance wavelength when the PDMS refractive index is 1.35 RIU (the temperature is 423.37 K). The fundamental mode loss of x-direction polarized light is 0.3443 dB/cm at 645 nm, and is almost negligible. This proves that the light pattern in x direction is hardly lost. Interestingly, the effective mode refractive index in the x and y directions is almost the same, while only the effective mode refractive index in the y-direction has an imaginary part.

### 3.2. Duty Ratios and Polish Depth

The lattice spacing Λ of PCF is set to 7.9 um, and the polishing depth h is set to 0.5Λ. The refractive index RI of the analyte varies with temperature. At temperatures of 348.15 K, 373.15 K, and 398.15 K, Figure 4a depicts the resonance wavelength and peak loss of the spectrum at various duty ratios.

As the duty ratio rises, the peak losses increase significantly. However, the resonant wavelength decreases slightly and gradually plateaus. This demonstrates that, with the exception of the extreme case in which the duty ratio is 0.2, the relationship between resonance changes and temperature change is roughly independent of duty ratios. According to the calculation method of amplitude sensitivity, the higher the duty ratio, the higher the amplitude sensitivity to temperature. Increasing the duty ratio can strengthen the coupling between the core guided mode and the SPP mode, resulting in greater energy dissipation in the sensing region. However, an excessively high duty ratio will lead to too much loss of the core mode, and in the experiment, the output light intensity detected by the spectrometer is too low, so a moderate duty ratio is adopted (d/Λ = 0.5).

Figure 4b represents the loss spectrum of y-polarized core guided mode at various temperatures and polishing depths. The polishing plane gradually moves away from the core region as the polishing depth h grows from 0.5Λ to Λ. In addition, the coupling between the y-polarized core mode and the SPP mode is gradually weakens, and the peak loss decreases. At the temperatures of 348.15 K, 373.15 K, and 398.15 K, when the polishing depth increases to Λ, the plasmon oscillations generated by the metal film on the polished surface resonates with the fundamental mode light wave. At these three temperatures, the resonance wavelengths are red-shifted with the temperature reduce. In general, when the temperature changes, the effect of the test substance PDMS on the resonance wavelength is much greater than that of the change in the dielectric constant in the gold film on the resonance wavelength. Nevertheless, with the temperature increases, the peak loss of the y-polarized core mode gradually decreases, and the coupling efficiency becomes weaker. With the same temperature change, the greater the polishing depth, the smaller the peak loss changes, that is, the greater the polishing depth, the lower the amplitude sensitivity of the sensor. Taking the amplitude sensitivity of the sensor and the manufacturing process into account, 0.5Λ is adopted as the polishing depth of the sensor.

### 3.3. Wavelength Sensitivity, Amplitude Sensitivity, and Corresponding Resolution

Amplitude sensitivity is one of the most crucial performance characteristics of optical fiber surface plasmon sensors. The following equation defines amplitude sensitivity generally [41].
(9)$$S_{A}=-\frac{1}{\alpha \left(\lambda ,n_{a}\right)}\frac{\partial \alpha \left(\lambda ,n_{a}\right)}{\partial n_{a}}\left(RIU^{-1}\right)$$
where $\alpha \left(\lambda ,n_{a}\right)$ stands for radiation loss and $\frac{\partial \alpha \left(\lambda ,n_{a}\right)}{\partial n_{a}}$ represents the variation of radiation loss with refractive index.

Similarly, wavelength sensitivity [41] is also a crucial parameter in fiber SPR sensing, and is defined as the difference between the resonant wavelength phase and the refractive index change in the measured material.
(10)$$S_{\lambda }=\Delta \lambda_{\mathrm{peak}\;}/\Delta n_{a}$$

The temperature of the substance to be tested in the sensor is adjusted so that its refractive index (RI) satisfies the refractive index of 1.350–1.380. As illustrated in Figure 5, when the refractive index of PDMS is 1.375 RIU, the amplitude sensitivity of the sensor can reach 376.76 RIU^−1^, the wavelength sensitivity of the sensor is 2164 nm/RIU and the temperature sensitivity of the sensor can reach 0.975 nm/K.

The SPR sensor resolution refers to the sensor’s ability to detect the lowest change in the quantity that can detect accurately. Therefore, the sensor resolution calculation [42] is also crucial to quantify the detection capability of the proposed sensor. It is defined as follows:(11)$$R\left(\mathrm{RIU}\right)=\Delta n_{a}\times \Delta \lambda_{\min}/\Delta \lambda_{\mathrm{peak}}$$
where $\Delta n_{a}=$ 0.005 RIU, $\Delta \lambda_{\min}=0.1\mathrm{nm}$, and $\Delta \lambda_{\mathrm{peak}}=14\mathrm{nm}$. When coated with 30 nm gold film, the device achieves a maximum resolution of 3.58 × 10^−5^ RIU (due to the change in with the temperature from 356.71 K to 423.37 K, the refractive index is changed from 1.35 to 1.38 in Figure 5.

### 3.4. Nano-Titania Thin Film Sensitization

Currently, titanium dioxide (${\mathrm{TiO}}_{2}$) has been widely studied for its chemical stability, optical, and photocatalytic properties. In this paper, titanium dioxide is used to sensitize and reduce the minimum resolution. A titanium dioxide layer is incorporated between the gold film and the polished surface to facilitate photoelectron transfer. The resonance wavelength and fundamental mode loss obtained after adding 30–70 nm ${\mathrm{TiO}}_{2}$ layers are illustrated in Figure 6. As the thickness of ${\mathrm{TiO}}_{2}$ layer increases, the resonance wavelength of SPR gradually redshifts and the fundamental mode loss gradually increases. When the thickness of ${\mathrm{TiO}}_{2}$ increases from 40 nm to 50 nm, the wavelength sensitivity increases from 3258.92 nm/RIU to 3392.86 nm/RIU. 

The sensitivity of ${\mathrm{TiO}}_{2}$ with a thickness of 60 nm and 70 nm is almost the same as that with a thickness of 50 nm, but at the same time, due to the fact that their fundamental mode loss is lower than that at 50 nm, the characterization of SPR phenomenon in the experiment will be affected, and its amplitude sensitivity will be reduced accordingly. Therefore, 50 nm thick titanium dioxide film is adopted for subsequent sensitization experiments.

In the original model, the sensitivity of the SPR sensor is 2164 nm/RIU when coated with 30 nm gold film, while the wavelength sensitivity reaches 3392.86 nm/RIU when sensitized by adding a titanium dioxide layer (as shown in Figure 7), and the temperature sensitivity of the sensor is increased to 1.320 nm/K. Assuming that the transmission intensity is 5%, the maximum resolution of this device reaches 1.47 × 10^−5^ RIU. Compared with the original device, the new device reduces the maximum resolution by 2.11 × 10^−5^ RIU. It can be seen that ${\mathrm{TiO}}_{2}$ increases the wavelength sensitivity of the sensor and reduces the maximum resolution of the device, thus greatly improving the sensor performance.

### 3.5. Mode Field Diameter

Mode Field Diameter (MFD) is used to characterize the distribution of the fundamental mode light in the core region of an optical fiber. In the core region, the strength of the fundamental mode is greatest at the axis and gradually declines with increasing distance from the axis. In the fiber, the optical energy is not entirely concentrated in the fiber core; rather, a portion of the energy is transmitted through the pores. The diameter of the fiber core cannot reflect the energy distribution within the core. Considering the fiber coupling, the larger the mode field diameter is, the higher the coupling efficiency is. 

The diameter of the mode field is defined as the maximum distance between two points at each of which the optical intensity is reduced to $1/\left(e^{2}\right)$ of the maximum optical intensity at the axis. The diameter of the mode field is expressed through the transmission power method as follows [43]:(12)$$MFD=2\sqrt{2\frac{\int^{\;}\left(r^{2}*poavz\right)}{\int^{\;}\left(poavz\right)}}$$
where poavz is the time-average power flow in the z direction. Figure 8 reveals the distribution of device mode field diameters at different wavelengths in the presence and absence of titanium dioxide. 

According to Figure 8, in the wavelength range of 700 nm to 900 nm, the diameter of the mode field increases progressively as the wavelength increases. The mode field diameter of the device can be improved by ${\mathrm{TiO}}_{2}$ film with a thickness of 50 nm. Compared with that in the near infrared band, the device with ${\mathrm{TiO}}_{2}$ film has a longer mode field diameter in the visible spectrum band (Table 2).

## 4. Conclusions

In summary, a D-shaped two-core PCF-based surface plasmon sensor with high sensitivity is introduced in this paper. Using the FEM method in COMSOL, the impacts of side polishing depth, pore duty ratio, and titanium dioxide sensing layer thickness on sensor performance are investigated. Theoretical calculation shows that when the thickness of the gold film is 30 nm, the higher temperature is, the lower the loss of the fundamental mode is, and the sensitivity of the sensor and the stability of the optical mode field are improved with the gradual increase in the duty ratio. Increasing the duty ratio simultaneously increases the sensitivity of the sensor and the stability of the optical mode field. With increasing polishing depth, the wavelength of resonance gradually moves to the red and the loss of fundamental mode steadily reduces. The peak loss of sensors working in intensity inquiry mode responds linearly to RI and temperature changes. Compared with other related works on the influence of temperature on SPR sensor, the highlights of this work can be summarized as follows. (1) The relationship between dielectric constant and thermal expansion of gold film as a function of temperature based on Drude model is established, and on this basis, the relationship between resonance wavelength in SPR sensor and device temperature, refractive index of substance to be measured, and other conditions is also studied. (2) This study provides a theoretical analysis basis for the preparation of other D-shaped sensors and PCF sensors, and can further promote the design of PCF sensors with higher sensitivity and better transmission performance.

## Figures and Tables

**Figure 1 materials-16-00037-f001:**
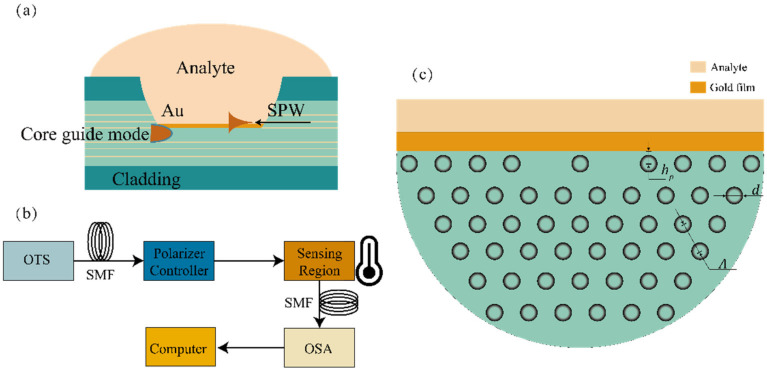
(**a**) Schematic diagram of coupling SPW with core mode in PCF waveguide. (**b**) Schematic diagram of experimental set-up for PCF-SPR based fiber optic sensor. (**c**) Cross-section of the D-shaped SPR sensor based on dual channel photonic crystal fiber.

**Figure 2 materials-16-00037-f002:**
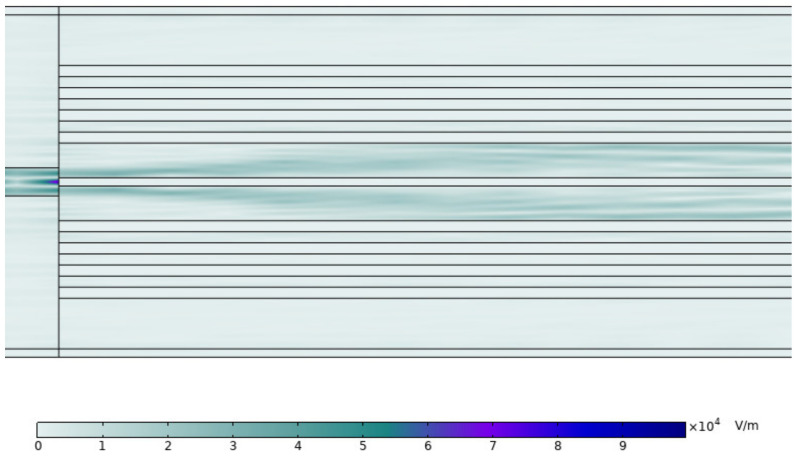
The electric field mode in the coupling region of PCF and SMF.

**Figure 3 materials-16-00037-f003:**
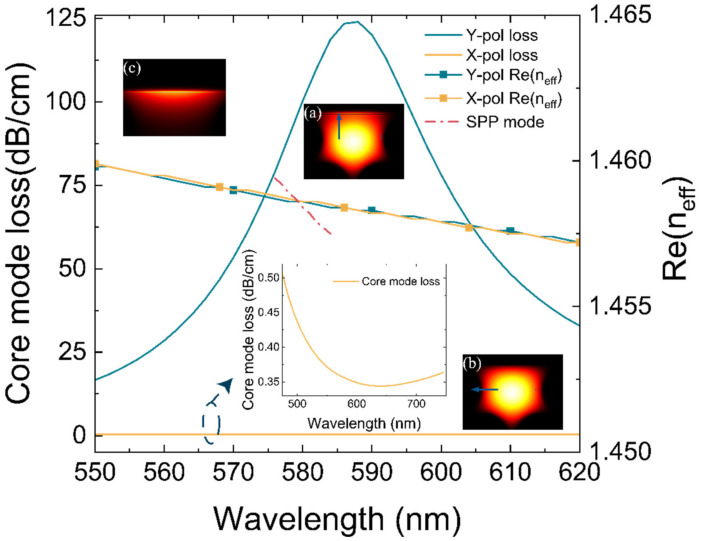
The fundamental mode loss (left axis) and the real part of the effective refractive index (right axis) of the x-polarized mode (yellow line) and y-polarized mode (green line) in the wavelength range of 550–620 nm, respectively. (**a**) Electric field mode distribution of y-polarized mode producing SPR effect. (**b**) Electric field distribution of x-polarized mode. (**c**) The SPP mode.

**Figure 4 materials-16-00037-f004:**
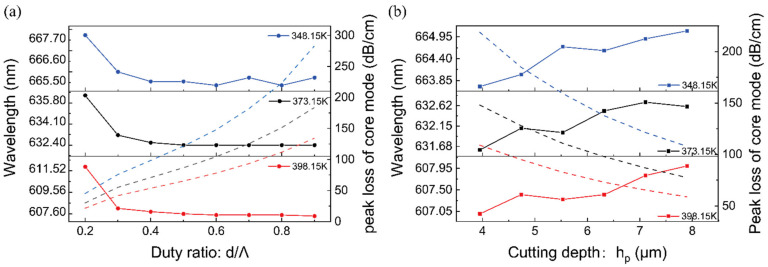
Dependence of the resonance wavelength (solid lines) and peak loss (dashed lines) at temperatures of 348.15 K, 373.15 K, and 398.15 K, (**a**) The effect of different duty cycle on resonance wavelength and fundamental mode loss. (**b**) The effect of different cutting depth on resonance wavelength and fundamental mode loss.

**Figure 5 materials-16-00037-f005:**
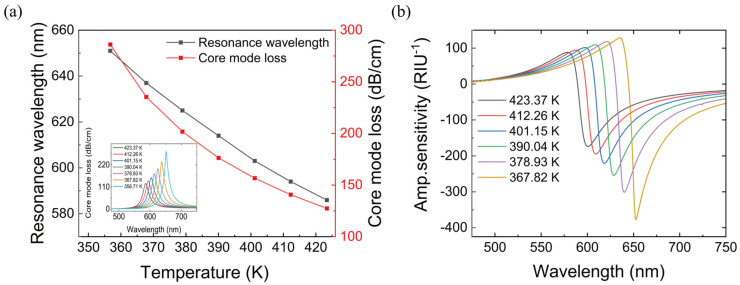
(**a**) Resonance wavelength and fundamental mode loss of the PDMS with the temperature of 356.71–423.37 K (refractive index of 1.350–1.380). (**b**) Sensor amplitude sensitivity at with the temperature of 367.82–423.37 K.

**Figure 6 materials-16-00037-f006:**
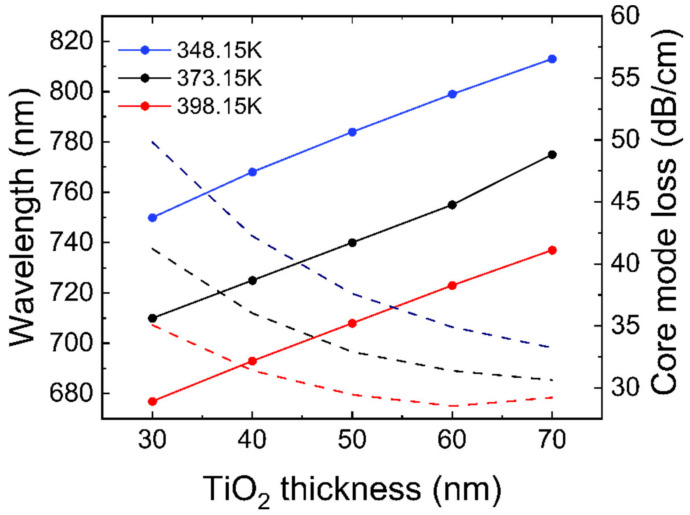
Effect of 30–70 nm thickness titanium dioxide on resonance wavelength (solid lines) and fundamental mode loss (dashed lines).

**Figure 7 materials-16-00037-f007:**
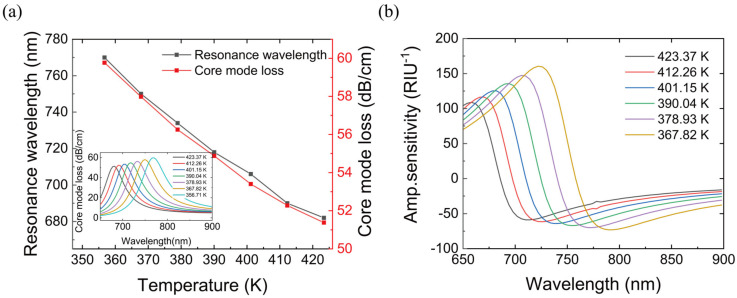
(**a**) After adding nano titanium dioxide layer for sensitization, resonance wavelength, and fundamental mode loss of the PDMS with a temperature of 356.71 K–423.37 K (refractive index of 1.350–1.380). (**b**) Sensor amplitude sensitivity with a temperature of 367.82 K–423.37 K.

**Figure 8 materials-16-00037-f008:**
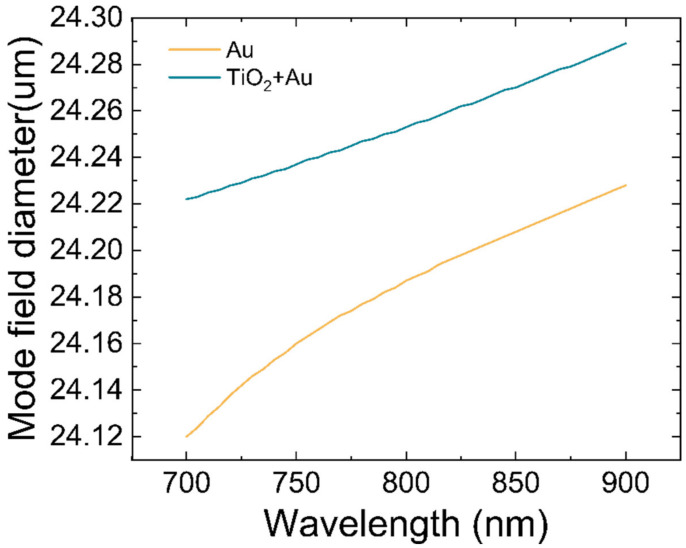
The mode field diameter of the sensor.

**Table 1 materials-16-00037-t001:** The parameters of the gold layer utilized in the computation.

Parameters	Value	Parameters	Value
Δ	0.77	*T_D_*	185 K
*Γ*	0.55	*E_F_*	5.51 eV
*γ*	1.42 × 10^−5^ K^−1^	*M*	0.44
*K_B_*	1.38 × 10^−23^ J/K		

**Table 2 materials-16-00037-t002:** Comparison of the properties of the proposed sensors with existing ones.

Ref.	Wavelength Sensitivity (nm/RIU)	Wavelength Resolution (RIU)	Amp. Sensitivity (RIU^−1^)
[44]	2000	5 × 10^−5^	200
[45]	2459	4.17 × 10^−5^	-
[46]	2900	3.44 × 10^−5^	120
[47]	~3000	2.67 × 10^−5^	-
[48]	3385	3.13 × 10^−5^	-
Proposed Sensor(Au)	2164	3.58 × 10^−5^	376.76
Proposed Sensor(with TiO_2_)	3392.86	1.47 × 10^−5^	73.12

## Data Availability

All the data are available in main text.

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
