# Peer review of "Simulation of High-Performance Surface Plasmon Resonance Sensor Based on D-Shaped Dual Channel Photonic Crystal Fiber for Temperature Sensing"

_materials, 2022, doi:10.3390/ma16010037_

Round 1

Reviewer 1 Report

This paper reports on the simulation of a temperature SPR sensor based on a D-shaped dual-channel photonic crystal fiber using the commercial COMSOL Multiphysics software package.

I would not recommend this paper for publication in its current form because it contains many grammatical errors and typos (see some examples below). Authors should take into account the comments below and check spelling carefully before resubmitting an article.

1.    Abstract, line 3 – replace “sen-sitizing” with “sensitizing”.

2.    Page 1, line 22 – insert space after “resonance”.

3.    Page 1, line 35 – insert space before “Note”.

4.    Page 2, line 42 – replace “bire-fringence” with “birefringence”.

5.    Page 3, Equation (1) – shift equation to the right.

6.    Page 3, line just after line 93 – insert space after “frequency”.

7.    Page 3, line 76 – “Stomatal diameter” is not a widely accepted term in photonics area, but rather in botany and biology sciences. Maybe it is better to use instead of “Stomatal diameter” – “pores’ diameter” or “the diameter of air holes”.

8.    Page 3, lines 1 and 2 after Eqn. (5) – replace “scaterring.[27]And” with  “scaterring [27]. And”.

9.    Page 5, Figure 3 caption – insert at the beginning of the caption – “Dependence of the resonance wavelength and peak loss at temperatures of 348.15K, 373.15K, 398.15K”.

10.              Page 7, line 160 – it is not clear what Figure number the authors refer to?

11.              Page 7, line 165 – insert space after “RIU”.

12.              Page 7, line 167 – replace “1.37” with “1.35”.

13.              Page 7, line 181 – insert space after “50nm”.

14.              Page 9, line 203 – use italic font for term “poavz”.

15.              Page 9, Table 2 caption – change caption to “Comparison of the properties of the proposed sensors with existing ones”.

16.              Page 9, line 212 – replace “comsol” with “Comsol”.

17.              Page 9, line 224 – replace “drude” with “Drude”.

18.              Page 10, Reference 18 – insert the authors’ names instead of “and,;; and.”

Reviewer 2 Report

Authors have presented a refractive index sensor based on side-polished D-shaped two-channel photonic crystal fiber (PCF) and surface plasmon resonance (SPR) for improved sensing performance. I have the following queries:

1. In the introduction the complete focus is on the SPR techniques and the authors claim to have improved sensitivity than conventional SPR sensors. How about the conventional Bloch Surface Wave technique, which shows much better sensitivity?  The authors need to widen the scope of the introduction section.

2. DPCF structure is not novel and is widely explored in literature. Thus, the authors need to add recent results in the introduction section.

3. Figure 4: Increasing the operating wavelength leads to an increase in the core mode loss, why?

4. Fir sensitivity, authors have considered only 1.35 - 1.38 refractive index variation only, which is very small for practical application. The authors need to provide the sensing results for 1.3 -  1.6 refractive index variation.

Reviewer 3 Report

Authors present simulation work on a D-Shape PCF structure. Detailed characterization of the performance of the sensor based on the design parameters is also presented. The subject is interesting for the fiber sensor community and the manuscript is well structured. However, in the opinion of this reviewer, the manuscript should clarify several issues before publication.

Minor comments:

1.- A thoughtful revision on the style is needed. Several typos are found. For example:

2.- Line 26: “field.Noble…” instead of “field. Noble…”

3.- SPWS in line 33 has not been defined previously.

4.- SPP in line 67 has not been defined previously.

5.-In Figure 1a the color used for Analyte is different than the one used in Fig 1b. Using the same color would help.

6.- Some edits on some figures would help to better understand the information of the plots. For instance, insets of Fig 2 and Fig 6a could have a bigger font size. Less (and probably integers) wavelength ticks in Fig 3 would help to understand faster the vertical axis.

7.- Reference [27] appears after a punctuation (line 93-94), and probably should appear before the punctuation.

8.- I recommend avoiding placing the figures just before a title or subtitle as in Fig.2 and Fig. 7

9.- Line 160 has a typo

10.- In line 181 one reads “nmthick” instead of “nm thick”; a space is needed.

11.- Typo in line 199

Major comments:

1.- The role of the two cores in the PCF structure is not clear. According to the authors, the structure of the PCF was selected “to be as simple and straightforward as possible” (lines 69-70). However, it is unclear what is the benefit of having two cores instead of one, which is always simpler. All the simulation results seem to correspond to one core and no information is found about a potential coupling of energy. Moreover, if the coupling is negligible, then authors should emphasize the reason behind using two cores instead of one. Based on Figure 1b, one can infer that the PCF in spliced between SMF segments. However, the authors avoid elaboration on the splice geometry of such coupling. Details on this important experimental feature needs to be included.

2.-The authors use PDMS to characterize the proposed structure. However, for a potential realization PDMS is not the best material to work with due to its intrinsic viscosity. It is not clear to this reviewer the use of this material in the simulations as the refractive index of other materials are also available (such as solvents). The reasoning of this should be clarify.

3.- If I understand correctly, this is a purely simulation work. If so, this should be pointed very clearly from the title and abstract.

4.- Although the subject is interesting, this approach (D shape-PCF-dual core-plasmon) has been reported previously. This reviewer has found, for instance:

https://doi.org/10.1007/s11468-022-01700-0

https://doi.org/10.1016/j.rinp.2019.102788

https://doi.org/10.1007/s11468-022-01637-4

10.1109/JPHOT.2022.3161468

https://doi.org/10.1016/j.ijleo.2020.164796

https://doi.org/10.1007/s11082-020-02555-7

Elaboration on the advantages and disadvantages of the proposed structure must be clearly incorporated in terms of the previously published work.

Round 2

Reviewer 3 Report

Authors have addressed all reviewer´s comments and concerns. The manuscript looks solid, Figures help to better understand the main ideas of the proposed sensor and in general the text is easier to read for non-experts.

A small typo in line 50 was found. A space is needed: Esteban Gonzalez.

Aside from that, this reviewer believes that the manuscript can be published in its present form.